# Natural Language Descriptions of Deep Visual Features

**Evan Hernandez**[1]    **Sarah Schwettmann**[1]    **David Bau**[1,2]    **Teona Bagashvili**[3]
**Antonio Torralba**[1]    **Jacob Andreas**[1]
[1]MIT CSAIL    [2]Northeastern University    [3]Allegheny College
{dez,schwett,teona,torralba,jda}@mit.edu    d.bau@northeastern.edu

## Abstract

Some neurons in deep networks specialize in recognizing highly specific perceptual, structural, or semantic features of inputs. In computer vision, techniques exist for identifying neurons that respond to individual concept categories like colors, textures, and object classes. But these techniques are limited in scope, labeling only a small subset of neurons and behaviors in any network. Is a richer characterization of neuron-level computation possible? We introduce a procedure (called MILAN, for **m**utual-**i**nformation-guided **l**inguistic **a**nnotation of **n**eurons) that automatically labels neurons with open-ended, compositional, natural language descriptions. Given a neuron, MILAN generates a description by searching for a natural language string that maximizes pointwise mutual information with the image regions in which the neuron is active. MILAN produces fine-grained descriptions that capture categorical, relational, and logical structure in learned features. These descriptions obtain high agreement with human-generated feature descriptions across a diverse set of model architectures and tasks, and can aid in understanding and controlling learned models. We highlight three applications of natural language neuron descriptions. First, we use MILAN for **analysis**, characterizing the distribution and importance of neurons selective for attribute, category, and relational information in vision models. Second, we use MILAN for **auditing**, surfacing neurons sensitive to human faces in datasets designed to obscure them. Finally, we use MILAN for **editing**, improving robustness in an image classifier by deleting neurons sensitive to text features spuriously correlated with class labels.[1]

## 1 Introduction

A surprising amount can be learned about the behavior of a deep network by understanding the individual neurons that make it up. Previous studies aimed at visualizing or automatically categorizing neurons have identified a range of interpretable functions across models and application domains: low-level convolutional units in image classifiers implement color detectors and Gabor filters (Erhan et al., 2009), while some later units activate for specific parts and object categories (Zeiler & Fergus, 2014; Bau et al., 2017). Single neurons have also been found to encode sentiment in language data (Radford et al., 2017) and biological function in computational chemistry (Preuer et al., 2019). Given a new model trained to perform a new task, can we automatically catalog these behaviors?

Techniques for characterizing the behavior of individual neurons are still quite limited. Approaches based on visualization (Zeiler & Fergus, 2014; Girshick et al., 2014; Karpathy et al., 2015; Mahendran & Vedaldi, 2015; Olah et al., 2017) leave much of the work of interpretation up to human users, and cannot be used for large-scale analysis. Existing automated labeling techniques (Bau et al., 2017; 2019; Mu & Andreas, 2020) require researchers to pre-define a fixed space of candidate neuron labels; they label only a subset of neurons in a given network and cannot be used to surface novel or unexpected behaviors.

This paper develops an alternative paradigm for labeling neurons with expressive, compositional, and open-ended annotations in the form of *natural language descriptions*. We focus on the visual

---

[1]Code, data, and an interactive demonstration may be found at `http://milan.csail.mit.edu/`.

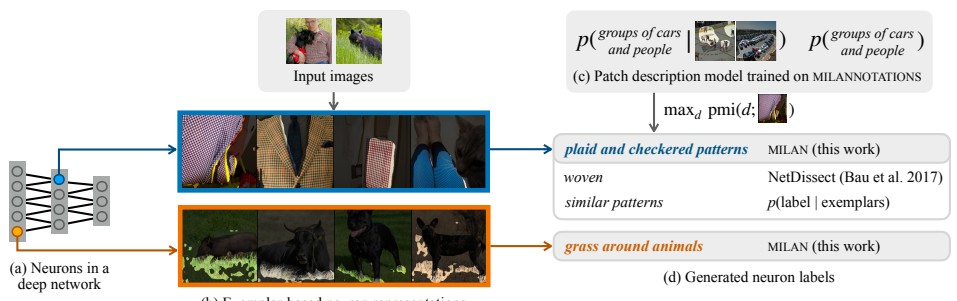

Figure 1: **(a)** We aim to generate natural language descriptions of individual neurons in deep networks. **(b)** We first represent each neuron via an *exemplar set* of input regions that activate it. **(c)** In parallel, we collect a dataset of fine-grained human descriptions of image regions, and use these to train a model of $p(\text{description} \mid \text{exemplars})$ and $p(\text{description})$. **(d)** Using these models, we search for a description that has high pointwise mutual information with the exemplars, ultimately generating highly specific neuron annotations.

domain: building on past work on information-theoretic approaches to model interpretability, we formulate neuron labeling as a problem of finding *informative* descriptions of a neuron's pattern of activation on input images. We describe a procedure (called MILAN, for **m**utual-**i**nformation-guided **l**inguistic **a**nnotation of **n**eurons) that labels individual neurons with fine-grained natural language descriptions by searching for descriptions that maximize pointwise mutual information with the image regions in which neurons are active. To do so, we first collect a new dataset of fine-grained image annotations (MILANNOTATIONS, Figure 1c), then use these to construct learned approximations to the distributions over image regions (Figure 1b) and descriptions. In some cases, MILAN surfaces neuron descriptions that more specific than the underlying training data (Figure 1d).

MILAN is largely model-agnostic and can surface descriptions for different classes of neurons, ranging from convolutional units in CNNs to fully connected units in vision transformers, even when the target network is trained on data that differs systematically from MILANNOTATIONS' images. These descriptions can in turn serve a diverse set of practical goals in model interpretability and dataset design. Our experiments highlight three: using MILAN-generated descriptions to (1) analyze the role and importance of different neuron classes in convolutional image classifiers, (2) audit models for demographically sensitive feature by comparing their features when trained on anonymized (blurred) and non-anonymized datasets, and (3) identify and mitigate the effects of spurious correlations with text features, improving classifier performance on adversarially distributed test sets. Taken together, these results show that fine-grained, automatic annotation of deep network models is both possible and practical: rich descriptions produced by automated annotation procedures can surface meaningful and actionable information about model behavior.

## 2 RELATED WORK

**Interpreting deep networks** MILAN builds on a long line of recent approaches aimed at explaining the behavior of deep networks by characterizing the function of individual neurons, either by visualizing the inputs they select for (Zeiler & Fergus, 2014; Girshick et al., 2014; Karpathy et al., 2015; Mahendran & Vedaldi, 2015; Olah et al., 2017) or by automatically categorizing them according to the concepts they recognize (Bau et al., 2017; 2018; Mu & Andreas, 2020; Morcos et al., 2018; Dalvi et al., 2019). Past approaches to automatic neuron labeling require fixed, pre-defined label sets; in computer vision, this has limited exploration to pre-selected object classes, parts, materials, and simple logical combinations of these concepts. While manual inspection of neurons has revealed that a wider range of features play an important role in visual recognition (e.g. orientation, illumination, and spatial relations; Cammarata et al. 2021) MILAN is the first automated approach that can identify such features at scale. Discrete categorization is also possible for *directions* in representation space (Kim et al., 2018; Andreas et al., 2017; Schwettmann et al., 2021) and for clusters of images induced by visual representations (Laina et al., 2020); in the latter, an off-the-shelf image captioning model is used to obtain language descriptions of the unifying visual concept for the cluster, although the descriptions miss low-level visual commonalities. As MILAN requires only a primitive procedure for generating model inputs maximally associated with the feature or direction of interest, future work might extend it to these settings as well.

**Natural language explanations of decisions**    Previous work aimed at explaining computer vision classifiers using natural language has focused on generating explanations for individual classification *decisions* (e.g., Hendricks et al., 2016; Park et al., 2018; Hendricks et al., 2018; Zellers et al., 2019). Outside of computer vision, several recent papers have proposed procedures for generating natural language explanations of decisions in text classification models (Zaidan & Eisner, 2008; Camburu et al., 2018; Rajani et al., 2019; Narang et al., 2020) and of *representations* in more general sequence modeling problems (Andreas & Klein, 2017). These approaches require task-specific datasets and often specialized training procedures, and do not assist with interpretability at the model level. To the best of our knowledge, MILAN is the first approach for generating compositional natural language descriptions for interpretability at the level of individual features rather than input-conditional decisions or representations. More fundamentally, MILAN can do so *independently* of the model being described, making it (as shown in Section 4) modular, portable, and to a limited extent task-agnostic.

## 3    APPROACH

**Neurons and exemplars**    Consider the neuron depicted in Figure 1b, located in a convlutional network trained to classify scenes (Zhou et al., 2017). When the images in Figure 1 are provided as input to the network, the neuron activates in *patches of grass near animals*, but not in grass without animals nearby. How might we automate the process of automatically generating such a description?

While the image regions depicted in Fig. 1b do not completely characterize the neuron's function in the broader network, past work has found that actionable information can be gleaned from such regions alone. Bau et al. (2020; 2019) use them to identify neurons that can trigger class predictions or generative synthesis of specific objects; Andreas & Klein (2017) use them to predict sequence outputs on novel inputs; Olah et al. (2018) and Mu & Andreas (2020) use them to identify adversarial vulnerabilities. Thus, building on this past work, our approach to neuron labeling also begins by representing each neuron via the set of input regions on which its activity exceeds a fixed threshold.

**Definition 1.** *Let $f : X \to Y$ be a neural network, and let $f_i(x)$ denote the activation value of the ith neuron in $f$ given an input $x$.[2] Then, an **exemplar representation** of the neuron $f_i$ is given by:*

$$E_i = \{x \in X : f_i(x) > \eta_i\} . \tag{1}$$

*for some threshold parameter $\eta_i$ (discussed in more detail below).*

**Exemplars and descriptions**    Given this explicit representation of $f_i$'s behavior, it remains to construct a **description** $d_i$ of the neuron. Past work (Bau et al., 2017; Andreas et al., 2017) begins with a fixed inventory of candidate descriptions (e.g. object categories), defines an exemplar set $E'_d$ for each such category (e.g. via the output of a semantic segmentation procedure) then labels neurons by optimizing $d_i := \arg\min_d \ \delta(E_i, E'_d)$ for some measure of set distance (e.g. Jaccard, 1912).

In this work, we instead adopt a probabilistic approach to neuron labeling. In computer vision applications, each $E_i$ is a set of image patches. Humans are adept at describing such patches (Rashtchian et al., 2010) and one straightforward possibility might be to directly optimize $d_i := \arg\max_d \ p(d \mid E_i)$. In practice, however, the distribution of human descriptions given images may not be well-aligned with the needs of model users. Fig. 2 includes examples of human-generated descriptions for exemplar sets. Many of them (e.g. *text* for AlexNet conv3-252) are accurate, but generic; in reality, the neuron responds specifically to text on screens. The generated description of a neuron should capture the specificity of its function—especially *relative to other neurons in the same model*.

We thus adopt an information-theoretic criterion for selecting descriptions: our final neuron description procedure optimizes pointwise mutual information between descriptions and exemplar sets:

**Definition 2.** *The **max-mutual-information description of the neuron** $f_i$ is given by:*

$$\text{MILAN}(f_i) := \arg\max_d \ \text{pmi}(d; E_i) = \arg\max_d \ \log p(d \mid E_i) - \log p(d) . \tag{2}$$

To turn Eq. (2) into a practical procedure for annotating neurons, three additional steps are required: constructing a tractable approximation to the exemplar set $E_i$ (Section 3.1), using human-generated image descriptions to model $p(d \mid E)$ and $p(d)$ (Section 3.2 and Section 3.3), and finding a high-quality description $d$ in the infinite space of natural language strings (Section 3.4).

---

[2]In this paper, we will be primarily concerned with neurons in convolutional layers; for each neuron, we will thus take the input space $X$ to be the space of all image patches equal in size to the neuron's receptive field.

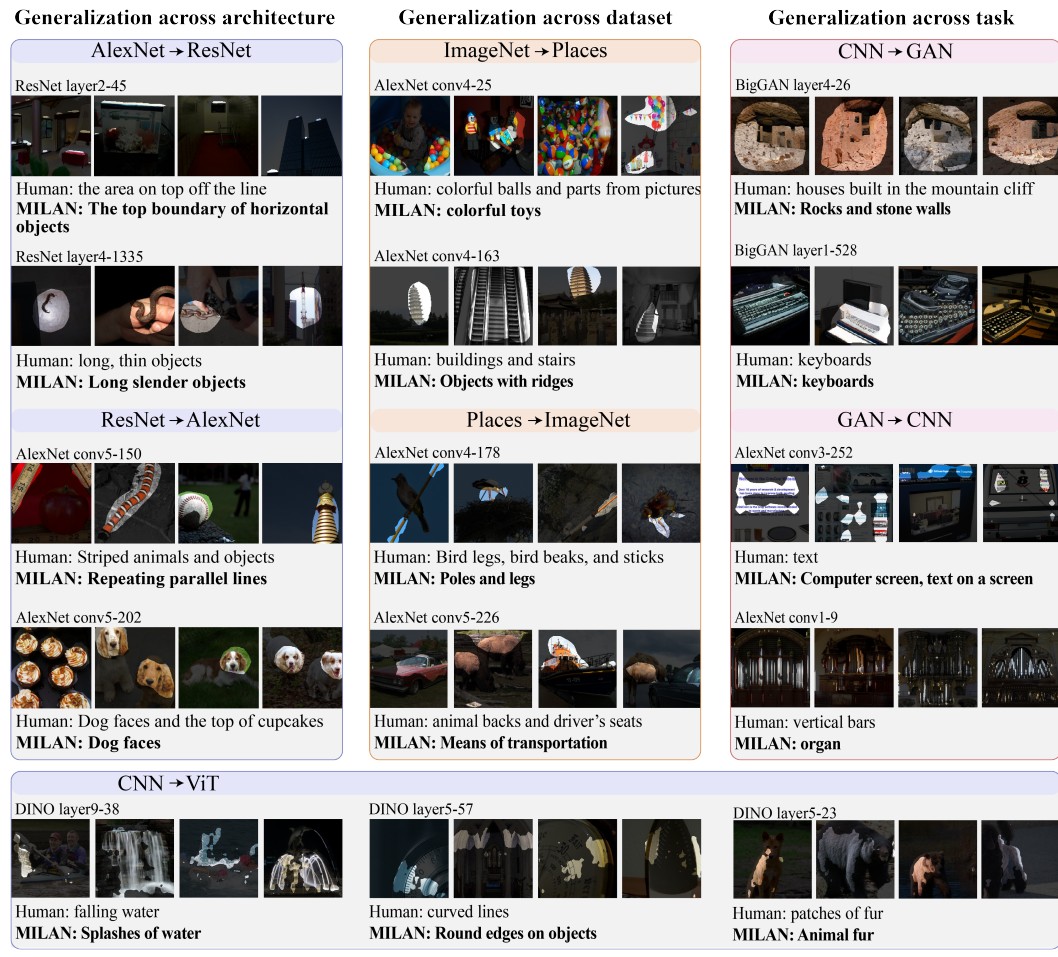

Figure 2: Examples of MILAN descriptions on the generalization tasks described in Section 4. Even highly specific labels (like *the top boundaries of horizontal objects*) can be predicted for neurons in new networks. Failure modes include semantic errors, e.g. MILAN misses the *cupcakes* in the *dog faces and cupcakes* neuron.

## 3.1 APPROXIMATING THE EXEMPLAR SET

As written, the exemplar set in Equation (1) captures a neuron's behavior on *all* image patches. This set is large (limited only by the precision used to represent individual pixel values), so we follow past work (Bau et al., 2017) by restricting each $E_i$ to the set of images that cause the greatest activation in the neuron $f_i$. For convolutional neurons in image processing tasks, sets $E_i$ ultimately comprise $k$ images with *activation masks* indicating the regions of those images in which $f_i$ fired (Fig. 1a; see Bau et al. 2017 for details). Throughout this paper, we use exemplar sets with $k = 15$ images and choose $\eta_i$ equal to the 0.99 percentile of activations for the neuron $f_i$.

## 3.2 MODELING $p(d \mid E)$ AND $p(d)$

The term $\mathrm{pmi}(d; E_i)$ in Equation (2) can be expressed in terms of two distributions: the probability $p(d \mid E_i)$ that a human would describe an image region with $d$, and the probability $p(d)$ that a human would use the description $d$ for any neuron. $p(d \mid E_i)$ is, roughly speaking, a distribution over *image captions* (Donahue et al., 2015). Here, however, the input to the model is not a single image but a set of image regions (the masks in Fig. 1a); we seek natural language descriptions of the common features of those regions. We approximate $p(d \mid E_i)$ with learned model—specifically the Show-Attend-Tell image description model of Xu et al. (2015) trained on the MILANNOTATIONS dataset described below, and with several modifications tailored to our use case. We approximate $p(d)$ with a two-layer LSTM language model (Hochreiter & Schmidhuber, 1997) trained on the text of MILANNOTATIONS. Details about both models are provided in Appendix B.

### 3.3 COLLECTING HUMAN ANNOTATIONS

As $p(d \mid E_i)$ and $p(d)$ are both estimated using learned models, they require training data. In particular, modeling $p(d \mid E_i)$ requires a dataset of captions that describe regions from multiple different images, such as the ones shown in Fig. 1. These descriptions must describe not only objects and actions, but all other details that individual neurons select for. Existing image captioning datasets, like MSCOCO (Lin et al., 2014) and Conceptual Captions (Sharma et al., 2018), only focus on scene-level details about a single image and do not provide suitable annotations for this task. We therefore collect a novel dataset of captions for image regions to train the models underlying MILAN.

First, we must obtain a set of image regions to annotate. To ensure that these regions have a similar distribution to the target neurons themselves, we derive them directly from the exemplar sets of neurons in a set of **seed models**. We obtain the exemplar sets for a subset of the units in each seed model in Table 1 using the method from Section 3.1. We then present each set to a human annotator and ask them to describe what is common to the image regions.

| Network | Arch. | Task | Datasets | Annotated | # Units |
|---|---|---|---|---|---|
| AlexNet | CNN | Class. | ImageNet Places365 | conv. 1–5 | 1152 1376 |
| ResNet152 | CNN | Class. | ImageNet Places365 | conv. 1 res. 1–4 | 3904 3904 |
| BigGAN | CNN | Gen. | ImageNet Places365 | res. 0–5 | 3744 4992 |
| DINO | ViT | BYOL | ImageNet | MLP 1–12 (first 100) | 1200 |

Table 1: Summary of MILANNOTATIONS, which labels 20k units across 7 models with different network architectures, datsasets, and tasks. Each unit is annotated by three human participants.

Table 1 summarizes the dataset, which we call MILANNOTATIONS. In total, we construct exemplar sets using neurons from seven vision models, totaling 20k neurons. These models include two architectures for supervised image classification, AlexNet (Krizhevsky et al., 2012) and ResNet152 (He et al., 2015); one architecture for image generation, BigGAN (Brock et al., 2018); and one for unsupervised representation learning trained with a "Bootsrap Your Own Latent" (BYOL) objective (Chen & He, 2020; Grill et al., 2020), DINO (Caron et al., 2021). These models cover two datasets, specifically ImageNet (Deng et al., 2009) and Places365 (Zhou et al., 2017), as well as two completely different families of models, CNNs and Vision Transformers (ViT) (Dosovitskiy et al., 2021). Each exemplar set is shown to three distinct human participants, resulting 60k total annotations. Examples are provided in Appendix A (Fig. 10). We recruit participants from Amazon Mechanical Turk. This data collection effort was approved by MIT's Committee on the Use of Humans as Experimental Subjects. To control for quality, workers were required to have a HIT acceptance rate of at least 95%, have at least 100 approved HITs, and pass a short qualification test. Full details about our data collection process and the collected data can be found in Appendix A.

### 3.4 SEARCHING IN THE SPACE OF DESCRIPTIONS

Directly decoding descriptions from $\mathrm{pmi}(d; E_i)$ tends to generate disfluent descriptions. This is because the $p(d)$ term inherently discourages common function words like *the* from appearing in descriptions. Past work language generation (Wang et al., 2020) has found that this can be remedied by first introducing a hyperparameter $\lambda$ to modulate the importance of $p(d)$ when computing PMI, giving a new **weighted PMI** objective:

$$\mathrm{wpmi}(d) = \log p(d \mid E_i) - \lambda \log p(d). \qquad (3)$$

Next, search is restricted to a set of captions that are high probability under $p(d \mid E_i)$, which are reranked according to Eq. (3). Specifically, we run beam search on $p(d \mid E_i)$, and use the full beam after the final search step as a set of candidate descriptions. For all experiments, we set $\lambda = .2$ and beam size to 50.

## 4 DOES MILAN GENERALIZE?

Because it is trained on a set of human-annotated exemplar sets obtained from a set of seed networks, MILAN is useful as an automated procedure only if it *generalizes* and correctly describes neurons in trained models with new architectures, new datasets, and new training objectives. Thus, before describing applications of MILAN to specific interpretability problems, we perform cross-

validation experiments within the MILANNOTA-
TIONS data to validate that MILAN can reliably label
new neurons. We additionally verify that MILAN
provides benefits over other neuron annotation
techniques by comparing its descriptions to three
baselines: NetDissect (Bau et al., 2017), which
assigns a single concept label to each neuron by
comparing the neuron's exemplars to semantic
segmentations of the same images; Compositional
Explanations (Mu & Andreas, 2020), which follows
a similar procedure to generate logical concept la-
bels; and ordinary image captioning (selecting de-
scriptions using $p(d \mid E)$ instead of $\text{pmi}(d; E)$).

| Model | CE | ND | $p(d \mid E)$ | $\text{pmi}(d; E)$ |
|---|---|---|---|---|
| AlexNet-ImageNet | .01 | .24 | .34 | **.38** |
| AlexNet-Places | .02 | .21 | .31 | **.37** |
| ResNet-ImageNet | .01 | .25 | .27 | **.35** |
| ResNet-Places | .03 | .22 | .30 | **.31** |

Table 2: BERTScores for neuron labeling meth-
ods relative to human annotations. MILAN ob-
tains higher agreement than Compositional Expla-
nations (CE) or NetDissect (ND).

**Method** In each experiment, we train MILAN on
a subset of MILANNOTATIONS and evaluate its per-
formance on a held-out subset. To compare MILAN
to the baselines, we train on all data except a single
held-out network; we obtain the baseline labels by
running the publicly available code with the default
settings on the held-out network. To test general-
ization within a network, we train on 90% of neu-
rons from each network and test on the remaining
10%. To test generalization across architectures, we
train on all AlexNet (ResNet) neurons and test on
all ResNet (AlexNet) neurons; we also train on all
CNN neurons and test on ViT neurons. To test gen-
eralization across datasets, we train on all neurons
from models trained on ImageNet (Places) and test
on neurons from models for the other datasets. To
test generalization across tasks, we train on all clas-

| Generalization | Train + Test | | BERTScore (f) |
|---|---|---|---|
| within network | AlexNet–ImageNet | | .39 |
| | AlexNet–Places | | .47 |
| | ResNet152–ImageNet | | .35 |
| | ResNet152–Places | | .28 |
| | BigGAN–ImageNet | | .49 |
| | BigGAN–Places | | .52 |
| | **Train** | **Test** | |
| across arch. | AlexNet | ResNet152 | .28 |
| | ResNet152 | AlexNet | .35 |
| | CNNs | ViT | .34 |
| across datasets | ImageNet | Places | .30 |
| | Places | ImageNet | .33 |
| across tasks | Classifiers | BigGAN | .34 |
| | BigGAN | Classifiers | .27 |

Table 3: BERTScores on held out neurons rela-
tive to the human annotations. Each train/test split
evaluates a different kind of generalization, ul-
timately evaluating how well MILAN generalizes
to networks with architectures, datasets, and tasks
unseen in the training annotations.

sifier neurons (GAN neurons) and test on all GAN neurons (classifier neurons). We measure perfor-
mance via BERTScore (Zhang et al., 2020) relative to the human annotations. Hyperparameters for
each of these experiments are in Appendix C.

**Results** Table 2 shows results for MILAN and all three baselines applied to four different net-
works. **MILAN obtains higher agreement with human annotations on held-out networks than
baselines.** It is able to surface highly specific behaviors in its descriptions, like the *splashes of water*
neuron shown in Figure 2 (*splashes* has no clear equivalent in the concept sets used by NetDissect
(ND) or Compositional Explanations (CE)). MILAN also outperforms the ablated $p(d \mid E)$ decoder,
justifying the choice of pmi as an objective for obtaining specific and high-quality descriptions.[3]

Table 3 shows that MILAN exhibits different degrees of generalization across models, with gener-
alization to new GAN neurons in the same network easiest and GAN-to-classifier generalization
hardest. **MILAN can generalize to novel architectures**. It correctly labels ViT neurons (in fully
connected layers) as often as it correctly labels other convolutional units (e.g., in AlexNet). We
observe that **transferability across tasks is asymmetric:** agreement scores are higher when trans-
ferring from classifier neurons to GAN neurons than the reverse. Finally, Figure 3 presents some
of MILAN's failure cases: when faced with new visual concepts, MILAN sometimes mislabels the
concept (e.g., by calling brass instruments *noodle dishes*), prefers a vague description (e.g., *similar
color patterns*), or ignores the highlighted regions and describes the context instead.

We emphasize that this section is primarily intended as a sanity check of the learned models underly-
ing MILAN, and not as direct evidence of its usefulness or reliability as a tool for interpretability. We

---

[3]It may seem surprising that ND outperforms CE, even though ND can only output one-word labels. One
reason is that ND obtains image segmentations from multiple *segmentation models*, which support a large vo-
cabulary of concepts. By contrast, CE uses a *fixed dataset* of segmentations and has a smaller base vocabulary.
CE also tends to generate complex formulas (with up to two logical connectives), which lowers its precision.

follow Vaughan & Wallach (2020) in arguing that the final test of any such tool must be its ability to produce actionable insights for human users, as in the three applications described below.

# 5 ANALYZING FEATURE IMPORTANCE

The previous section shows that MILAN can generalize to new architectures, datasets, and tasks. The remainder of this paper focuses on applications that *use* generated labels to understand how neurons influence model behavior. As a first example: descriptions in Figure 2 reveal that neurons have different degrees of specificity. Some neurons detect objects with spatial constraints (*the area on top of the line*), while others fire for low-level but highly specific perceptual qualities (*long, thin objects*). Still others detect perceptually similar but fundamentally different objects (*dog faces and cupcakes*). How important are these different classes of neurons to model behavior?

**Method** We use MILAN trained on all convolutional units in MILANNOTATIONS to annotate every neuron in ResNet18-ImageNet. We then score each neuron according to one of seven criteria that capture different *syntactic* or *structural* properties of the caption. Four **syntactic** criteria each count the number of times that a specific part of speech appears in a caption: *nouns*, *verbs*, *prepositions*, and *adjectives*. Three **structural** criteria measure properties of the entire caption: its *length*, the *depth* of its parse tree (a rough measure of its compositional complexity, obtained from the spaCy parser of Honnibal et al. 2020), and its *maximum word difference* (a measure of the semantic coherence of the description, measured as the maximum Euclidean distance between any two caption words, again obtained via spaCy). Finally, neurons are incrementally ablated in order of their score. The network is tested on the ImageNet validation set and its accuracy recorded. This procedure is then repeated, deleting 2% of neurons at each step. We also include five trials in which neurons are ordered *randomly*. Further details and examples of ablated neurons are provided in Appendix D.

**Results** Figure 4 plots accuracy on the ImageNet validation set as a function of the number of ablated neurons. Linguistic features of neuron descriptions highlight several important differences between neurons. First, **neurons captioned with many adjectives or prepositions (that is, neurons that capture attributes and relational features) are relatively important to model behavior.** Ablating these neurons causes a rapid decline in performance compared to ablating random neurons or nouns. Second, **neurons that detect dissimilar concepts appear to be less important.** When the

**MILAN failures**

AlexNet conv5-239
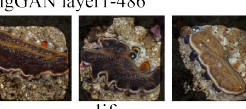

Human: yellow and green animals, food, instruments, and objects
**MILAN: Noodle dishes**

BigGAN layer1-486
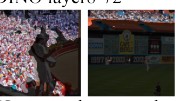

Human: sea life
**MILAN: Similar color patterns**

DINO layer8-72
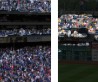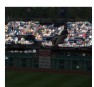

Human: the crowd
**MILAN: Athletes**

Figure 3: Examples of MILAN failures. Failure modes include incorrect generalization (**top**), vague descriptions for concepts not seen in the training set (**middle**), and mistaking the context for the highlighted regions (**bottom**).

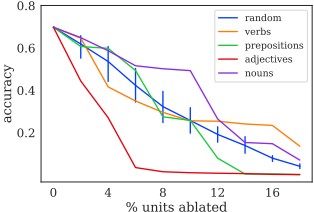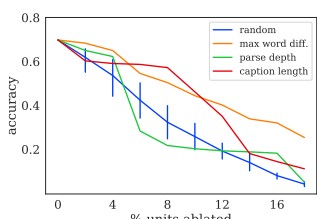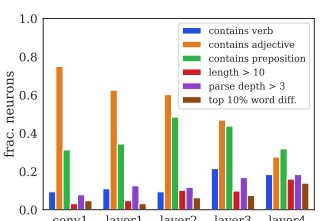

Figure 4: ResNet18 accuracy on the ImageNet validation set as units are ablated (**left, middle**), and distribution of neurons matching syntactic and structural criteria in each layer (**right**). In each configuration, neurons are scored according to a property of their generated description (e.g., number of nouns/words in description, etc.), sorted based on their score, and ablated in that order. Neurons described with adjectives appear crucial for good performance, while neurons described with very different words (measured by word embedding difference; *max word diff.*) appear less important for good performance. Adjective-selective neurons are most prevalent in early layers, while neurons with large semantic differences are more prevalent in late ones.

caption contains highly dissimilar words (*max word diff.*), ablation hurts performance substantially less than ablating random neurons. Such neurons sometimes detect non-semantic compositions of concepts like the *dog faces and cupcakes* neuron shown in Fig. 2; Mu & Andreas (2020) find that these units contribute to non-robust model behavior. We reproduce their robustness experiments using these neurons in Section 5 (Figure 14) and reach similar conclusions. Finally, Figure 4 highlights that neurons satisfying each criterion are not evenly distributed across layers—for example, **middle layers contain the largest fraction of relation-selective neurons** measured via prepositions.

## 6    AUDITING ANONYMIZED MODELS

One recent line of work in computer vision aims to construct *privacy-aware* datasets, e.g. by detecting and blurring all faces to avoid leakage of information about specific individuals into trained models (Yang et al., 2021). But to what extent does this form of anonymization actually reduce models' reliance on images of humans? We wish to understand if models trained on blurred data still construct features that can human faces, or even specific categories of faces. A core function of tools for interpretable machine learning is to enable auditing of trained models for such behavior; here, we apply MILAN to investigate the effect of blurring-based dataset privacy.

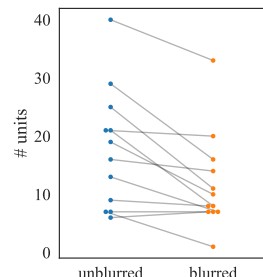

**Method**    We use MILAN to caption a subset of convolutional units in 12 different models pretrained for image classification on the blurred ImageNet images (*blurred* models). These models are distributed by the original authors of the blurred ImageNet dataset (Yang et al., 2021). We caption the same units in models pretrained on regular ImageNet (*unblurred* models) obtained from torchvision (Paszke et al., 2019). We then manually inspect all neurons in the blurred and unblurred models for which MILAN descriptions contain the words *face*, *head*, *nose*, *eyes*, and *mouth* (using exemplar sets containing only unblurred images).

Figure 5: Change in # of face neurons found by MILAN (each pair of points is one model architecture). Blurring reduces, but does not eliminate, units selective for unblurred faces.

**Results**    Across models trained on ordinary ImageNet, MILAN identified 213 neurons selective for human faces. Across models trained on blurred ImageNet, MILAN identified 142 neurons selective for human faces. **MILAN can distinguish between models trained on blurred and unblurred data** (Fig. 5). However, it also reveals that **models trained on blurred data acquire neurons selective for unblurred faces**. Indeed, it is possible to use MILAN's labels to extract these face-selective neurons directly. Doing so reveals that several of them are not simply face detectors, but appear to selectively identify female faces (Fig. 6b) and Asian faces (Fig. 6c). Blurring does not prevent models from extracting highly specific features for these attributes. Our results in this section highlight the use of MILAN for both quantitative and qualitative, human-in-the loop auditing of model behavior.

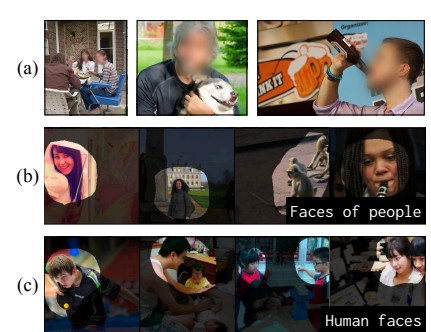

Figure 6: **(a)** The blurred ImageNet dataset. **(b–c)** Exemplar sets and labels for two neurons in a blurred model that activate on unblurred faces—and appear to preferentially (but not exclusively) respond to faces in specific demographic categories.

## 7    EDITING SPURIOUS FEATURES

**Spurious correlations** between features and labels are a persistent problem in machine learning applications, especially in the presence of mismatches between training and testing data (Storkey, 2009). In object recognition, one frequent example is correlation between backgrounds and objects (e.g. cows are more likely to appear with green grass in the background, while fish are more likely to appear with a blue background; Xiao et al. 2020). In a more recent example, models trained on joint text and image data are subject to "text-based adversarial attacks", in which e.g. an apple with the word *iPod* written on it is classified as an iPod (Goh et al., 2021). Our final experiment shows that MILAN can be used to reduce models' sensitivity to these spurious features.

**Data** We create a controlled dataset imitating Goh et al. (2021)'s spurious text features. The dataset consists of 10 ImageNet classes. In the training split, there are 1000 images per class; 500 are annotated with (correct) text labels in the top-left corner. The test set contains 100 images per class (from the ImageNet validation set); in all these images, a *random* (usually incorrect) text label is included. We train and evaluate a fresh ResNet18 model on this dataset, holding out 10% of the training data as a validation dataset for early stopping. Training details can be found in Appendix E.

**Method** We use MILAN to obtain descriptions of every residual neuron in the model as well as the first convolutional layer. We identify all neurons whose description contains *text*, *word*, or *letter*. To identify spurious neurons, we first assign each text neuron an independent *importance score* by removing it from the network and measuring the resulting drop in validation accuracy (with non-adversarial images). We then sort neurons by importance score (with the least important first), and successively ablate them from the model.

**Results** The result of this procedure on adversarial test accuracy is shown in Fig. 8. Training on the spurious data substantially reduces ResNet18's performance on the adversarial test set: the model achieves 58.8% accuracy, as opposed to 69.9% when tested on non-spurious data. MILAN identifies 300 text-related convolutional units (out of 1024 examined) in the model, confirming that the model has indeed devoted substantial capacity to identifying text labels in the image. Figure 7c shows an example neurons specifically selective for *airline* and *truck* text. By deleting only 13 such neurons, test accuracy is improved by 4.9% (a 12% reduction in overall error rate).[4] This increase cannot be explained by the sorting procedure described above: if instead we sort *all* neurons according to validation accuracy (orange line), accuracy improves by less than 1%. Thus, while this experiment does not completely eliminate the model's reliance on text features, it shows that MILAN**'s predictions enable direct editing of networks to partially mitigate sensitivity to spurious feature correlations.**

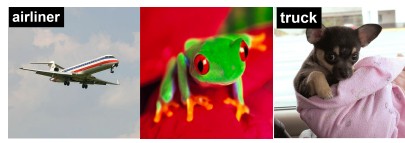

(a) training dataset     (b) adversarial test dataset

layer3-134, "words and letters"

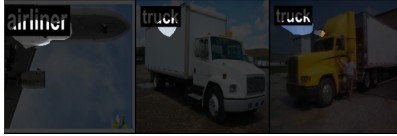

(c) text neuron

Figure 7: Network editing. **(a)** We train an image classifier on a synthetic dataset in which half the images include the class label written in text in the corner. **(b)** We evaluate the classifier on an adversarial test set, in which every image has a *random* textual label. **(c)** Nearly a third of neurons in the trained model model detect text, hurting its performance on the test set.

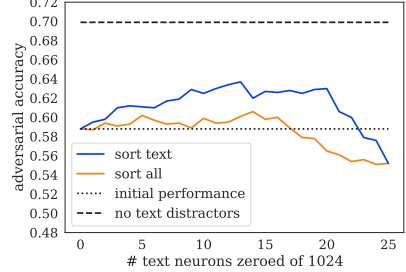

Figure 8: ResNet18 accuracy on the adversarial test set as neurons are incrementally ablated. Neurons are sorted by the model's validation accuracy when that single neuron is ablated, then ablated in that order. When ablating neurons that select for the spurious text, the accuracy improves by 4.9 points. When zeroing arbitrary neurons, accuracy still improves, but by much less.

## 8 CONCLUSIONS

We have presented MILAN, an approach for automatically labeling neurons with natural language descriptions of their behavior. MILAN selects these descriptions by maximizing pointwise mutual information with image regions in which each neuron is active. These mutual information estimates are in turn produced by a pair of learned models trained on MILANNOTATIONS, a dataset of fine-grained image annotations released with this paper. Descriptions generated by MILAN surface diverse aspects of model behavior, and can serve as a foundation for numerous analysis, auditing, and editing techniques workflows for users of deep network models.

---

[4]Stopping criteria are discussed more in Appendix E; if no adversarial data is used to determine the number of neurons to prune, an improvement of 3.1% is still achievable.

IMPACT STATEMENT

In contrast to most past work on neuron labeling, MILAN generates neuron labels using another black-box learned model trained on human annotations of visual concepts. With this increase in expressive power come a number of potential limitations: exemplar-based explanations have known shortcomings (Bolukbasi et al., 2021), human annotations of exemplar sets may be noisy, and the captioning model may itself behave in unexpected ways far outside the training domain. The MI-LANNOTATIONS dataset was collected with annotator tests to address potential data quality issues, and our evaluation in Section 4 characterizes prediction quality on new networks; we nevertheless emphasize that these descriptions are *partial* and potentially *noisy* characterizations of neuron function via their behavior on a fixed-sized set of representative inputs. MILAN complements, rather than replaces, both formal verification (Dathathri et al., 2020) and careful review of predictions and datasets by expert humans (Gebru et al., 2018; Mitchell et al., 2019).

ACKNOWLEDGMENTS

We thank Ekin Akyürek and Tianxing He for helpful feedback on early drafts of the paper. We also thank IBM for the donation of the Satori supercomputer that enabled training BigGAN on MIT Places. This work was partially supported by the MIT-IBM Watson AI lab, the SystemsThatLearn initiative at MIT, a Sony Faculty Innovation Award, DARPA SAIL-ON HR0011-20-C-0022, and a hardware gift from NVIDIA under the NVAIL grant program.

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

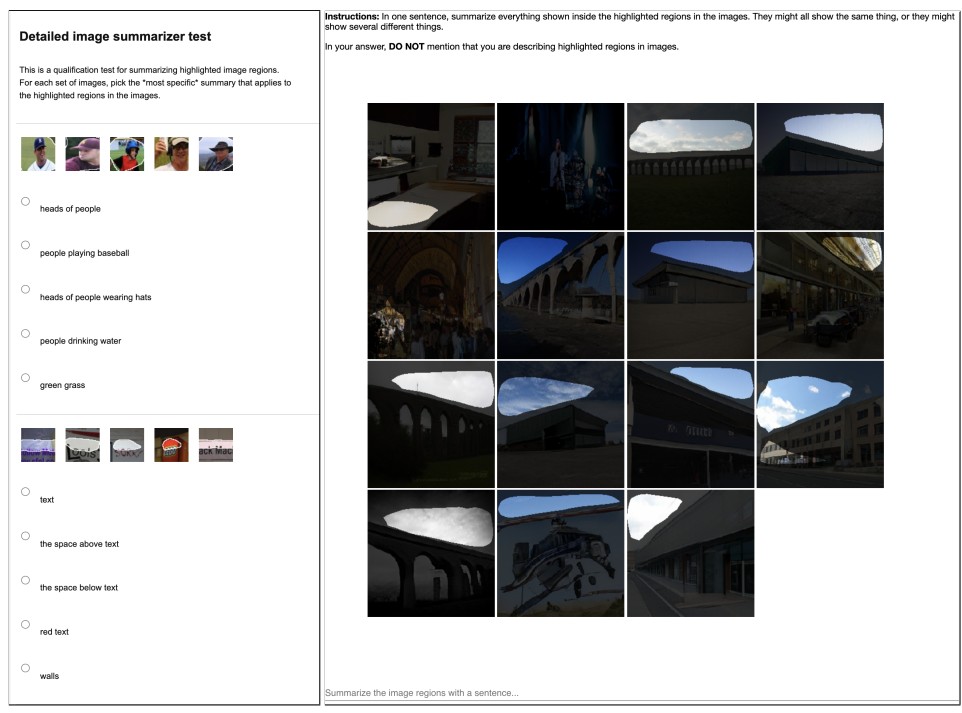

(a) qualification test  (b) annotation form

Figure 9: Screenshots of the Amazon Mechanical Turk forms we used to collect the CaNCAn dataset. **(a)** The qualification test. Workers are asked to pick the best description for two hand-chosen neurons from a model not included in our corpus. **(b)** The annotation form. Workers are shown the top-15 highest-activating images for a neuron and asked to describe what is common to them in one sentence.

Tianyi Zhang, Varsha Kishore, Felix Wu, Kilian Q. Weinberger, and Yoav Artzi. Bertscore: Evaluating text generation with bert. In *International Conference on Learning Representations*, 2020. URL https://openreview.net/forum?id=SkeHuCVFDr.

Bolei Zhou, Agata Lapedriza, Aditya Khosla, Aude Oliva, and Antonio Torralba. Places: A 10 million image database for scene recognition. *IEEE transactions on pattern analysis and machine intelligence*, 2017.

Bolei Zhou, Hang Zhao, Xavier Puig, Tete Xiao, Sanja Fidler, Adela Barriuso, and Antonio Torralba. Semantic understanding of scenes through the ade20k dataset. *International Journal of Computer Vision*, 127(3):302–321, 2019.

## A MILANNOTATIONS

We recruited annotators from Amazon Mechanical Turk to describe one neuron at a time given its top-activating images. A screenshot of the template is shown in Figure 9b. Participants were given the instructions:

> **Instructions:** In one sentence, summarize everything shown inside the highlighted regions in the images. They might all show the same thing, or they might show several different things.
> In your answer, **DO NOT** mention that you are describing highlighted regions in images.

Workers were given up to an hour to complete each annotation, but early trials revealed they required about 30 seconds per HIT. We paid workers $0.08 per annotation, which at $9.60 per hour exceeds the United States federal minimum wage.

Figure 10: Example human annotations for neuron exemplars in MILANNOTATIONS, which contains annotations for neurons in seven networks. Each set of images is annotated by three distinct human participants.

To control for quality, we required workers to pass a short qualification test in which they had to choose the most descriptive caption for two manually chosen neurons from VGG-16 (Simonyan & Zisserman, 2015) trained on ImageNet (not included as part of MILANNOTATIONS). A screenshot of this test is shown in Figure 9a.

Table 4 shows the inter-annotator agreement of neuron annotations for each model, and Table 5 shows some corpus statistics broken down by model and layer. Layers closest to the image (early layers in CNNs and later layers in GANs) are generally described with more adjectives than other layers, while annotations for layers farther from the image include more nouns, perhaps highlighting the low-level perceptual role of the former and the scene- and object-centric behavior of the latter. Layers farther from the image tend to have longer descriptions (e.g. in BigGAN-ImageNet, AlexNet-ImageNet), but this trend is not consistent across all models (e.g. in models trained on Places365,

| Model | Dataset | IAA |
|---|---|---|
| AlexNet | ImageNet | .25 |
|  | Places365 | .27 |
| ResNet152 | ImageNet | .21 |
|  | Places365 | .17 |
| BigGAN | ImageNet | .26 |
|  | Places365 | .24 |
| DINO | ImageNet | .23 |

Table 4: Average inter-annotator agreement among human annotations, measured in BERTScore. Some models have clearer neuron exemplars than others.

the middle layers have the longest average caption length).

# B   MILAN IMPLEMENTATION DETAILS

## B.1   IMPLEMENTING $p(d \mid E)$

We build on the Show, Attend, and Tell (SAT) model for describing images (Xu et al., 2015). SAT is designed for describing the high-level content of a single images, so we must make several modifications to support our use case, where our goal is to describe *sets* of *regions* in images.

| Model | Layer | # Units | # Words | Len. | % Noun | % Adj | % Prep |
|---|---|---|---|---|---|---|---|
| AlexNet-ImageNet | conv1 | 64 | 185 | 4.8 | 37.5 | 24.3 | 12.2 |
| | conv2 | 192 | 384 | 5.5 | 37.8 | 19.4 | 13.2 |
| | conv3 | 384 | 661 | 5.3 | 41.0 | 16.4 | 13.0 |
| | conv4 | 256 | 608 | 5.5 | 43.1 | 11.9 | 12.5 |
| | conv5 | 256 | 693 | 5.5 | 46.0 | 10.2 | 10.4 |
| AlexNet-Places365 | conv1 | 96 | 153 | 4.3 | 38.4 | 26.8 | 12.7 |
| | conv2 | 256 | 297 | 4.8 | 37.8 | 26.0 | 12.7 |
| | conv3 | 384 | 412 | 4.7 | 40.2 | 24.8 | 10.5 |
| | conv4 | 384 | 483 | 4.4 | 43.7 | 19.9 | 10.3 |
| | conv5 | 256 | 486 | 4.1 | 45.8 | 17.6 | 10.6 |
| ResNet152-ImageNet | conv1 | 64 | 285 | 4.7 | 43.8 | 11.8 | 10.3 |
| | layer1 | 256 | 653 | 5.5 | 43.1 | 10.5 | 12.5 |
| | layer2 | 512 | 936 | 5.1 | 44.0 | 12.7 | 12.6 |
| | layer3 | 1024 | 1222 | 4.2 | 49.6 | 10.9 | 11.3 |
| | layer4 | 2048 | 1728 | 4.6 | 47.8 | 8.6 | 7.8 |
| ResNet152-Places365 | conv1 | 64 | 283 | 5.2 | 47.3 | 11.1 | 14.6 |
| | layer1 | 256 | 633 | 5.3 | 46.3 | 9.4 | 13.3 |
| | layer2 | 512 | 986 | 5.8 | 46.0 | 8.3 | 13.8 |
| | layer3 | 1024 | 1389 | 4.8 | 48.2 | 6.7 | 12.7 |
| | layer4 | 2048 | 1970 | 5.3 | 46.3 | 5.5 | 11.9 |
| BigGAN-ImageNet | layer0 | 1536 | 1147 | 3.9 | 52.4 | 7.8 | 8.2 |
| | layer1 | 768 | 853 | 3.5 | 53.0 | 9.4 | 8.9 |
| | layer2 | 768 | 618 | 3.2 | 52.6 | 12.3 | 9.5 |
| | layer3 | 384 | 495 | 3.7 | 49.9 | 14.3 | 10.9 |
| | layer4 | 192 | 269 | 3.3 | 47.9 | 18.0 | 13.4 |
| | layer5 | 96 | 69 | 2.6 | 53.6 | 22.8 | 14.6 |
| BigGAN-Places365 | layer0 | 2048 | 1062 | 4.2 | 53.3 | 5.4 | 8.3 |
| | layer1 | 1024 | 708 | 3.9 | 55.0 | 6.1 | 11.5 |
| | layer2 | 1024 | 410 | 4.6 | 52.7 | 8.1 | 16.3 |
| | layer3 | 512 | 273 | 5.2 | 50.4 | 7.6 | 15.0 |
| | layer4 | 256 | 192 | 4.6 | 47.5 | 9.3 | 14.9 |
| | layer5 | 128 | 123 | 4.2 | 46.7 | 13.5 | 13.0 |
| DINO-ImageNet | layer0 | 100 | 320 | 4.4 | 45.7 | 12.7 | 4.8 |
| | layer1 | 100 | 321 | 4.2 | 49.8 | 9.1 | 6.8 |
| | layer2 | 100 | 285 | 3.9 | 53.3 | 6.2 | 7.5 |
| | layer3 | 100 | 312 | 3.9 | 54.4 | 6.2 | 7.1 |
| | layer4 | 100 | 304 | 3.9 | 53.5 | 4.4 | 7.0 |
| | layer5 | 100 | 287 | 3.5 | 55.1 | 5.5 | 5.2 |
| | layer6 | 100 | 377 | 3.9 | 51.3 | 8.2 | 5.4 |
| | layer7 | 100 | 374 | 3.8 | 52.0 | 6.4 | 6.2 |
| | layer8 | 100 | 330 | 3.4 | 53.0 | 7.0 | 8.8 |
| | layer9 | 100 | 350 | 3.1 | 56.1 | 6.3 | 9.6 |
| | layer10 | 100 | 369 | 3.9 | 50.3 | 9.3 | 8.2 |
| | layer11 | 100 | 294 | 3.3 | 52.4 | 7.5 | 9.4 |
| **Total** | | 20272 | 4597 | 4.5 | 48.7 | 9.4 | 10.9 |

Table 5: Corpus statistics for MILANNOTATIONS descriptions broken down by model and layer. The **# Words** column reports the number of unique words used across all layer annotations, the **Len.** column reports the average number of words in each caption for that layer, and the **%** columns report the percentage of all words across all captions for that layer that are a specific part of speech.

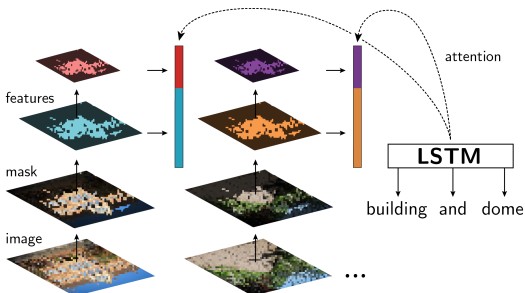

Figure 11: Neuron captioning model. Given the set of top-activating images for a neuron and masks for the regions of greatest activation, we extract features maps from each convolutional layer of a pretrained image classifier. We then downsample the masks and use them to pool the features before concatenating them into a single feature vector per image. These feature vectors are used as input to the decoder attention mechanism.

In the original SAT architecture, a single input image $x$ is first converted to visual features by passing it through an encoder network $g$, typically an image classifier pretrained on a large dataset. The output of the last convolutional layer is extracted as a matrix of visual features:

$$v = [v_1; v_2; \ldots; v_k]$$

These visual features are passed to a decoder LSTM whose hidden state is initialized as a function of the mean of the visual features $\overline{v} = 1/k \sum_i v_i$. At each time step, the decoder attends over the features using an additive attention mechanism (Bahdanau et al., 2015), then consumes the attenuated visual features and previous token as input to predict the next token.

The SAT architecture makes few assumptions about the structure of the visual features. We will take advantage of this generality and modify how $v$ is constructed to support our task, leaving the decoder architecture intact.

Now, instead of a single image $x$, the model inputs are the $k$ top-activating images $x_j$ for a neuron as well as a mask $m_j$ for each image that highlights the regions of greatest activation. Our task is to describe what the neuron is detecting, based strictly on the highlighted regions of the $x_j$. In support of this, the visual features must (1) include information about all $k$ images, (2) encode multiple resolutions of the images to capture both low-level perceptual and high-level scene details about the image, and (3) pay most (but not exclusive) attention to the regions of greatest activation in the image.

**Describing sets of images**   The $k$ features in SAT correspond to different spatial localities of a single image. In our architecture, each feature $v_j$ corresponds to one input image $x_j$.

**Encoding multiple resolutions**   Instead of encoding the image with just the last convolutional layer of $g$, we use pooled convolutional features from every layer. Formally, let $g_\ell(x)$ denote the output of layer $\ell$ in the pretrained image encoder with $L$ layers, and let pool denote a pooling function that uses the mask to pool the features (described further below). The feature vector for the $j$th image $x_j$ is:

$$v_j = \left[ \text{pool}(m_j, g_1(x_j)) ; \; \ldots \; ; \text{pool}(m_i, g_L(x_j)) \right]$$

**Highlighting regions of greatest activation**   Each of the top-activating images $x_j$ that we hand to our model comes with a mask $m_j$ highlighting the image regions of greatest activation. We incorporate these masks into the pooling function pool from above. Specifically, we first downsample the mask $m_j$ to the same spatial shape as $g_\ell(x_j)$ using bilinear interpolation, which we denote upsample($m_j$). We then apply the mask to each channel $c$ at layer $\ell$, written $g_{\ell,c}(x_j)$, via element-wise multiplication ($\odot$) with upsample($m_j$). Finally, we sum spatially along each channel, resulting in a length $c$ vector. Formally:

$$\text{pool}_c(g_\ell(x_j)) = \mathbb{1}^\top \text{vec}(\text{upsample}(m_j) \odot g_{\ell,c}(x_j))$$

Each $v_i$ is thus a length $\sum_\ell C_\ell$ vector, where $C_\ell$ is the number of channels at layer $\ell$ of $g$.

| Gen. | Train + Test | | # Units | # Words | Len. | % Noun | % Adj | % Prep |
|------|--------------|--|---------|---------|------|--------|-------|--------|
| within netwok | AlexNet–ImageNet | | 115 | 100 | 3.5 | 45.7 | 16.4 | 11.9 |
| | AlexNet–Places | | 137 | 46 | 2.5 | 49.3 | 28.7 | 9.6 |
| | ResNet–ImageNet | | 390 | 121 | 2.8 | 52.2 | 23.8 | 11.7 |
| | ResNet–Places | | 390 | 376 | 4.3 | 46.5 | 8.7 | 10.9 |
| | BigGAN–ImageNet | | 374 | 112 | 2.2 | 59.8 | 17.5 | 10.4 |
| | BigGAN–Places | | 499 | 245 | 3.8 | 54.2 | 6.0 | 9.0 |
| | **Train** | **Test** | | | | | | |
| across arch. | AlexNet | ResNet | 7808 | 326 | 3.0 | 46.1 | 21.0 | 8.9 |
| | ResNet | AlexNet | 2528 | 275 | 2.7 | 48.0 | 27.1 | 6.4 |
| | CNNs | ViT | 1200 | 200 | 2.6 | 55.0 | 18.2 | 13.0 |
| across dataset | ImageNet | Places | 10272 | 271 | 2.2 | 58.8 | 14.0 | 13.8 |
| | Places | ImageNet | 8800 | 309 | 3.1 | 47.8 | 26.9 | 7.8 |
| across task | Classifiers | BigGAN | 8736 | 202 | 2.1 | 53.0 | 25.3 | 6.1 |
| | BigGAN | Classifiers | 10336 | 336 | 3.2 | 54.3 | 14.2 | 16.8 |
| **Total** | | | 51585 | 1002 | 2.7 | 51.9 | 19.8 | 11.1 |

Table 6: Statistics for MILAN-generated descriptions on the held-out neurons from the generalization experiments of Section 4. Columns are the same as in Table 5.

Throughout our experiments, $g$ is a ResNet101 pretrained for image classification on ImageNet, provided by PyTorch Paszke et al. (2019). We extract visual features from the first convolutional layer and all four residual layers. We do not fine tune any parameters in the encoder. The decoder is a single LSTM cell with an input embedding size of 128 and a hidden size of 512. The attention mechanism linearly maps the current hidden state and all visual feature vectors to size 512 vectors before computing attention weights. We always decode for a maximum of 15 steps. The rest of the decoder is exactly the same as in Xu et al. (2015).

The model is trained to minimize cross entropy on the training set using the AdamW optimizer Loshchilov & Hutter (2019) with a learning rate of 1e-3 and minibatches of size 64. We include the double stochasticity regularization term used by Xu et al. (2015) with $\lambda = 1$. We also apply dropout ($p = .5$) to the hidden state before predicting the next word. Across configurations, 10% of the training data is held out and used as a validation set, and training stops when the model's BLEU score (Papineni et al., 2002) does not improve on this set for 4 epochs, up to a maximum of 100 epochs.

## B.2    IMPLEMENTING $p(d)$

We implement $p(d)$ using a two-layer LSTM language model (Hochreiter & Schmidhuber, 1997). We use an input embedding size of 128 with a hidden state size and cell size of 512. We apply dropout to non-recurrent connections ($p = .5$) during training and hold out 10% of the training dataset as a validation set and following the same early stopping procedure as in Appendix B.1, except we stop on validation loss instead of BLEU.

## C    GENERALIZATION EXPERIMENT DETAILS

In each experiment, MILAN is trained with the hyperparameters described in Appendix B and Section 3.4, with the sole exception being the within-network splits—for these, we increase the early stopping criterion to require 10 epochs of no improvement to account for the training instability caused by the small training set size.

To obtain NetDissect labels, we obtain image exemplars with the same settings as we do for MILAN, and we obtain segmentations using the full segmentation vocabulary minus the textures.

To obtain Compositional Explanations labels, we search for up to length 3 formulas (comprised of not, and, and or operators) with a beam size of 5 and no length penalty. Image region exemplars and corresponding segmentations come from the ADE20k dataset (Zhou et al., 2019).

Finally, Table 6 shows statistics for MILAN descriptions generated on the held out sets from each generalization experiment. Compared to human annotators (Table 5), MILAN descriptions are on

Figure 12: Randomly chosen examples of MILAN-generated descriptions from the generalization experiments of Section 4.

average shorter (2.7 vs. 4.5 tokens), use fewer unique words (1k vs. 4.6k), and contain adjectives twice as often (9.4% vs. 19.8%). Figure 12 contains additional examples, chosen at random.

## D    ANALYSIS EXPERIMENT DETAILS

We obtain the ResNet18 model pretrained on ImageNet from `torchvision` (Paszke et al., 2019). We obtain neuron descriptions for the same layers that we annotate in ResNet152 (Section 3.3) using the MILAN hyperparameters described in Section 3.2 and Section 3.4. We obtain part of speech tags, parse trees, and word vectors for each description from spaCy (Honnibal et al., 2020).

Figure 13 shows examples of neurons that scored high under each criterion (and consequently were among the first ablated in Fig. 5). Note that these examples include some failure cases of MILAN: for example, in the **# verbs** example, MILAN incorrectly categorizes all brass instruments as flutes; and

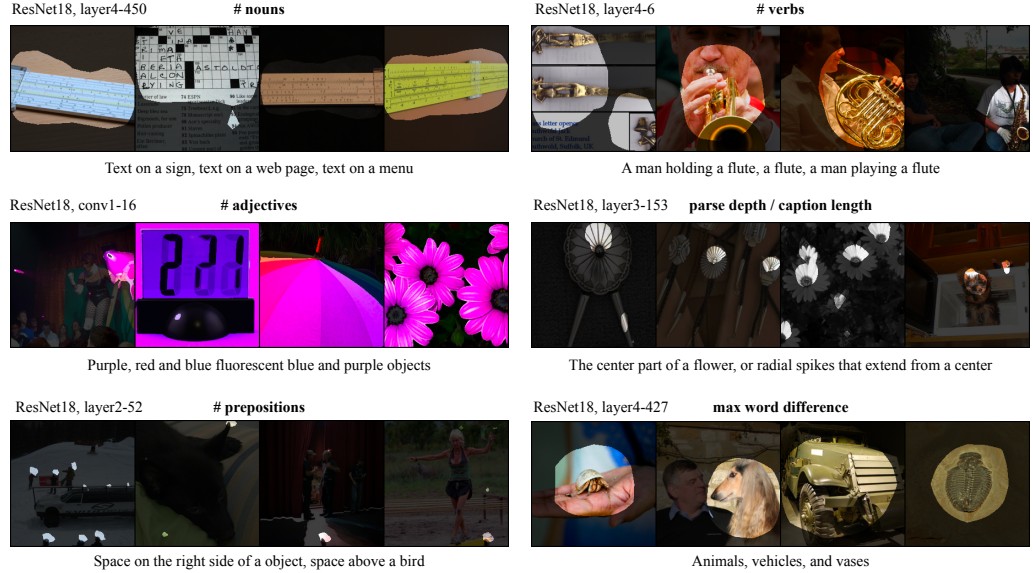

Figure 13: Examples of ablated neurons for each condition Section 5, chosen from among the first 10 ablated.

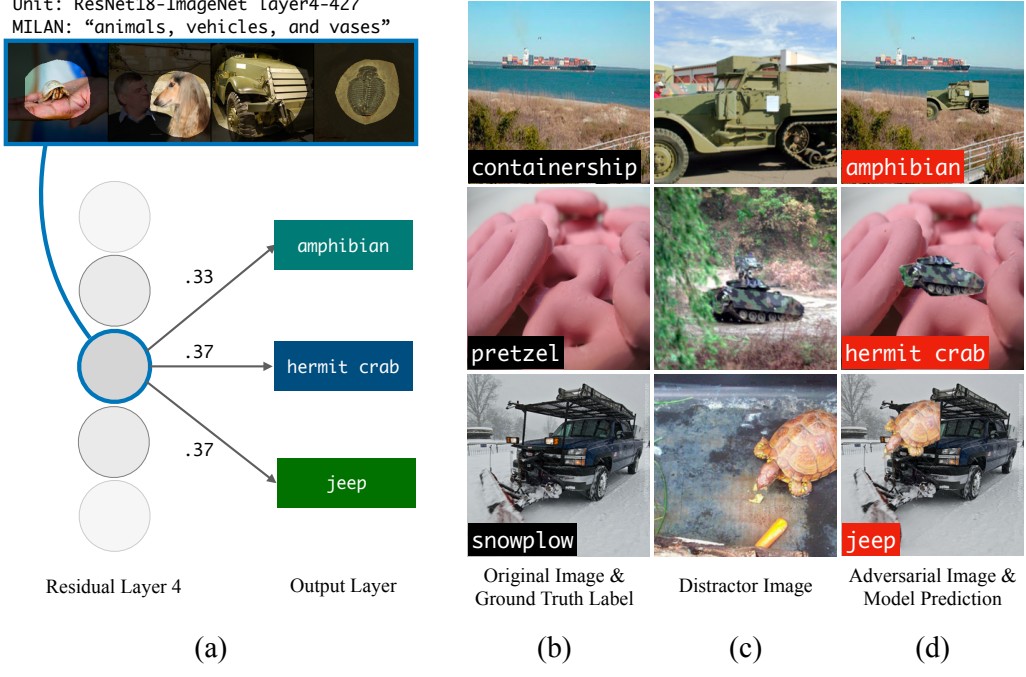

Figure 14: Cut-and-paste adversarial attacks highlighting non-robust behavior by a neuron that scored high on the *max-word-diff* criterion of Section 5. **(a)** MILAN finds this neuron automatically because the generated description mentions two or more dissimilar concepts: *animals* and *vehicles*. The neuron is directly connected to the final fully-connected output layer, and strongly influences *amphibian*, *hermit crab*, and *jeep* predictions according to the connection weights. **(b)** To construct adversarial inputs, we pick three images from the ImageNet validation set that do not include concepts detected by the neuron. **(c)** We then select a different set of images to act as distractors that do include the concepts detected by the neuron. **(d)** By cutting and pasting the central object from the distractor to the original image, the model is fooled into predicting a class label that is completely unrelated to the pasted object: e.g., it predicts *amphibian* when the military vehicle is pasted.

in the **# adjectives** example, the description is disfluent. Nevertheless, these examples confirm our intuitions about the kinds of neurons selected for by each scoring criterion, as described in Section 5.

We hypothesized in Section 5 that neurons scoring high on the *max-word-diff* criterion correspond to non-robust behavior by the model. Figure 14 provides some evidence for this hypothesis: we construct cut-and-paste adversarial inputs in the style of Mu & Andreas (2020). Specifically, we look at the example *max-word-diff* neuron shown in Figure 13, crudely copy and paste one of the objects mentioned in its description (e.g., a *vehicle*-related object like a half track), and show that this can cause the model to predict one of the other concepts in the description (e.g., an *animal*-related class like *amphibian*).

## E    EDITING EXPERIMENT DETAILS

**Hyperparameters**    We train a randomly initialized ResNet18 on the spurious training dataset for a maximum of 100 epochs with a learning rate of 1e-4 and a minibatch size of 128. We annotate the same convolutional and residual units we did for ResNet152 in Section 3.3. We stop training when validation loss does not improve for 4 epochs.

**How many neurons should we remove?**    In practice, we cannot incrementally test our model on an adversarial set. So how do we decide on the number of neurons to zero? One option is to look solely at validation accuracy. Figure 15 recreates Figure 8 with accuracy on the held out validation set (which is distributed like the training dataset) instead of accuracy on the adversarial test set. The accuracy starts peaks and starts decreasing earlier than in Fig. 8, but if we were to choose the number to be the *largest before validation accuracy permanently decreases*, we would choose 8 neurons, which would still result in a 3.1% increase in adversarial accuracy.

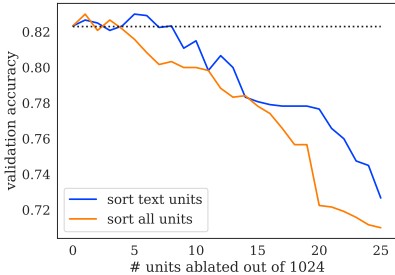

Figure 15: Same as Fig. 8, but shows accuracy on the *validation* dataset, which is distributed identically to the training dataset. Dotted line denotes initial accuracy.

