# OpenReview forum: "Natural Language Descriptions of Deep Visual Features"
_ICLR.cc/2022/Conference — ICLR 2022 Oral_

### Official Review · Reviewer_YviU · 2021-11-01

**Correctness:** 3
**Technical Novelty And Significance:** 3
**Empirical Novelty And Significance:** 3
**Recommendation:** 8
**Confidence:** 4

**Main Review:**

I want to preface my review by stating that I am a domain expert in NLP and not CV. While I attempted a preliminary search for related work, I am not sure if there is prior work in specifically generating language descriptions of neurons (I understand that there is prior work in explaining neuron behaviour by examining inputs that activate it). Hence I may significantly modify my review based on the response of my peer reviewers.

This is a strong piece of work that clearly states its hypotheses and carefully designs experiments to test said hypotheses. The technical novel of the particular method is low, however I do not think that is the point of this work. What this paper does show is a novel way to interpret model behaviour, and allows for very useful downstream applications (e.g. fast filtering of neurons activated by a particular feature through text selection). I advocate for acceptance of this work, and propose some potential improvement. I do have some concerns about the scoping of this work. The title and description suggests a more general method, however experiments are purely based on images. I think the paper title should be more finely scoped (NL descriptions of Deep Image Features).

Major:
- Do these transfer results generalize to other tasks? or are they specific to image models? Or are they specific to the fact that all of these image models are trained/evaluated on (two) datasets that are similar to each other in terms of image distribution? I think perhaps the paper makes this technique seem more portable than it actually is. For example I would have to collect data for my task distribution in order for this technique to work. If you can show this working with pretrained zero-shot image captioning models than this result would be a lot more convincing.
- How do image captioning models do on the task of generating descriptions conditioned on inputs? The implicit hypothesis here is that they do not work well because they operate on higher levels of abstraction, but I would like to see empirical results of this by using caption models as baselines (e.g. zero-shot, fine-tuned).
- What do generated descriptions look like? How long are the descriptions? What is the vocabulary size? How diverse are the descriptions?
- What does the released dataset look like? Do you control for bias in the description? Diversity? Coverage?
- This objective not only needs to maximize over all input space but also all description space. The former is talked about but the latter is skimmed over (by stating that beam search is performed). It's unclear whether beam search would yield useful description if the description is more complex (e.g. in a language task like question-answering).
- It would be nice to show spurious feature filtering results on something that is not a contrived task (e.g. classification with spurious text labels on top left corner of image) and instead on something impactful (e.g. removing racial/gender bias by filtering out neurons with corresponding descriptions).


Minor:
- typo; convlutional network

Questions:
- Why is there so much variance across architecture pairs? For example large gains on AlexNet -> Places and ResNet -> ImageNet but small gains on ResNet -> Places and AlexNet -> ImageNet.

**Summary Of The Paper:**

This work proposes a framework for interpreting model behaviour by generating language descriptions of neurons in the model. The method is trained to maximize mutual information between the description and input examples that activate the neuron. The experiments show results for image classification, generation, and unsupervised representation learning. In particular, this method yields model description that achieve higher BERTScore than max-likelihood training when transferring to neurons in new architectures. Moreover the authors show how this technique can be used to compute correlations between descriptions and neuron importance (e.g. wrt classification accuracy), interpreting model behaviour (e.g. anonymized models still select for unblurred faces), and that one can remove spurious feature corrleations by removing models with specific descriptions.

**Summary Of The Review:**

Strong paper demonstrating how to generation descriptions of image features and how to use method to analyze model behaviour and filter out neurons by description.

---

> ### Author Response · Authors · 2021-11-16
> **Response to R3 (1/2)**
>
> Thank you for the comments and suggestions!
>
> **The scope of the work should be clarified.** We agree, and we’ll change the paper title to “Natural Language Descriptions of Deep Visual Features” to reflect the finer scope. We designed MILAN to be general enough that it could be applied to other classes of models--e.g. NLP models--but since this paper focuses on vision models, we will make that explicit in the title.
>
> **Do these results transfer to (non-vision) tasks?** Our experiments focus on vision models, so we do not want to make over-broad conclusions about how MILAN will generalize to non-vision tasks. That said, we think this is a promising area for future research! Questions to answer (for applying MILAN to e.g. NLP models) include: (1) how can we obtain good exemplar sets for these neurons, and (2) what kinds of concepts would we expect to see in these models? (e.g., what is the analogue of a “perceptual feature” in language?)
>
> **What about transferring to more diverse vision datasets?** We’ve shown some evidence  that MILAN can generalize to neurons trained on new datasets (Section 4), but there are certainly limitations. We wouldn’t expect MILAN trained on ImageNet/Places365 data to generalize to, e.g., models trained on medical images. In such a setting, MILAN would likely still be able to describe low-level perceptual features (edges, spatial relations, etc.), but not ones detecting high-level medical concepts. We think that domain-specific gaps like this one could be closed by collecting additional data for domain-specific concepts like those in medical image analysis.
>
> We chose ImageNet/Places365 because these datasets contain a diverse array of concepts that are relevant to many vision tasks (object detection, semantic segmentation, scene classification, etc.). The datasets are similar to each other insofar as they both contain objects and people in scenes; however, the object- and scene-level content differs widely (e.g., ImageNet contains 125 distinct kinds of dogs, while Places365 contains no dog-specific classes but 3 separate categories of parking lots). The fact that annotation models trained on the Places365-specific subset of MILANNOTATIONS generalize to ImageNet indicates that some degree of transfer is already possible.
>
> **How do off-the-shelf image captioning models work on this task?** We agree this would be an interesting baseline. We’re working on applying a SAT model trained on COCO; we will add it to the appendix in the next version we upload.
>
> **What are some stats on generated descriptions?** Good question. We are working on adding a table to the appendix that is similar to Table 5, but for descriptions generated on a held-out model.
>
> **What does the released dataset look like?** Table 5 in the appendix provides a high-level overview of collected data for different models. We realize this is missing dataset-wide statistics; we’ll update it to include that. We’ll also add a large set of random examples from the dataset to the appendix to give a sense for what people say.
>
> **Did you control for bias in the dataset?** We did not implement any special controls for bias. To the extent that the exemplar image regions were taken from the data that models were trained on, MILANNOTATIONS inherits whatever biases are implicitly present in the underlying datasets (ImageNet/Places). We anticipate that MILANNOTATIONS contains some biases in how annotators choose to describe people--for example, annotators sometimes seem to refer to large groups of light-skinned people as “people,” while specifically using specific racial terms for other groups. Other biases (that are different in their nature) include a tendency for annotators to focus on entities in the highlighted regions of the images, instead of the visual commonalities between the regions. Some work attempts to mitigate the former kinds of biases by introducing adversarial losses to models trained on the data or by augmenting the underlying dataset [e.g., 1, 2, 5]; we think this line of work is complementary to ours, and could be used to reduce bias in future iterations of MILAN/MILANNOTATIONS.
>
> **How do you search over the space of neuron descriptions?** We realize this is a little unclear in the text, and we’re working on adding some additional context. To answer the question: we use beam search to obtain a set of high-probability candidate descriptions by taking the whole beam after the final step of the search, and then we rerank the descriptions in that set according to their PMI with the exemplars. We chose beam search because it is a relatively standard search procedure in image captioning, VQA, and other sequence generation tasks. Alternative search procedures include top-k or top-p sampling, but these have been shown to favor more disfluent descriptions [3, 4]. We will add an explicit numbered algorithm describing the procedure to the final version of the paper.

---

> > ### Author Response · Authors · 2021-11-16
> > **Response to R3 (2/2)**
> >
> > **Why not use MILAN to fix more impactful spurious correlations, e.g. gender or racial bias?** We agree this would be interesting to explore! Our fixing experiments were intended as a proof of concept that these kinds of interventions can have measurable and controllable effects on the underlying model. We do believe MILAN has the potential to reduce model sensitivity to spurious features in real-world scenarios. However, past work [e.g. 1, 2] has shown that proper handling of gender/racial bias requires substantial care, so we think these experiments belong in a separate paper. We hope to do this in future work.
> >
> > *References*
> >
> > [1] Hendricks et al. Women Also Snowboard: Overcoming Bias in Captioning Models. 2018
> > [2] Wang et al. Balanced Datasets Are Not Enough: Estimating and Mitigating Gender Bias in Deep Image Representations. 2019.
> > [3] Welleck et al. Neural Text Generation with Unlikelihood Training. 2019.
> > [4] Welleck et al. Consistency of a Recurrent Language Model with Respect to Incomplete Decoding. 2020.
> > [5] Zhang et al. Mitigating Unwanted Biases with Adversarial Learning

---

### Official Review · Reviewer_eaMc · 2021-11-02

**Correctness:** 4
**Technical Novelty And Significance:** 3
**Empirical Novelty And Significance:** 4
**Recommendation:** 8
**Confidence:** 4

**Main Review:**

The paper is about visualization and explainability. It is good read and inspirational to me since I have been an advocate of the sub-field. To be able go deeper on making use of the intermediate stages of a network, for visibility as well as new AI product features. The summary above shows the short of the discoveries. I have to experiment with the technique myself to go further. However, it looks promising.

I also appreciate the amount of effort put on testing the system, on many architectures, datasets, and tasks. If there is more, I would like to have better characterization of the limits of the approach.

Grammars, typos, etc.:
pg.1, par.1: convlution —> convolution


**Summary Of The Paper:**

The authors introduced MILAN, for mutual-information-guided linguistic annotation of neurons) that automatically labels neurons with open-ended, compositional, natural language descriptions.
This is done by searching for descriptions that maximize point-wise mutual information with the image regions in which the neurons are active.

It uses (Bau, et. al. 2017) model for the selection of 15 images with regions, and Xu et. al. 2015 (Show-Attend-Tell) with a modification in pmi (probability of mutual information) for describing these regions. The Show-Attend-Tell model is trained on MILANANOTATIONS dataset, a large contribution of the technique. It is a dataset of images and fine-grained region descriptions. The dataset is comprised of 20K neurons (sets of regions) with descriptions.

TESTING:
The testing section is a large part of the contribution as well.
* Section 4: MILAN obtained higher agreement with human annotations on held-out networks than baseline. It also shows that the model works across architecture, dataset, and task
* Section 5: neurons captioned with many adjectives or prepositions are relatively important to model behavior
* Section 6: Models trained on blurred faces acquire neurons selective for blurred faces
* Section 7: Networks devotes substantial capacity to identifying text labels in images.


**Summary Of The Review:**

* A higher level of explainability of the regions responsible for network final result
* A thorough analysis of the actionable insights that can be used for the model
* Limits of the approach needs to be discussed further

---

> ### Author Response · Authors · 2021-11-16
> **Response to R2**
>
> Thank you for the comments.
>
> **The limits of the approach should be discussed further.** We agree--we tried to capture some of the limitations with Figure 3, at the end of Section 4, and in the Ethics Statement. We will expand on these in the final version and will include some discussion of Figure 3 in the main text.

---

> > ### Comment · Reviewer_eaMc · 2021-11-23
> > **checked the added examples in Figure 12**
> >
> > thank you. Nothing further from me.

---

### Official Review · Reviewer_CDo5 · 2021-11-06

**Correctness:** 4
**Technical Novelty And Significance:** 3
**Empirical Novelty And Significance:** 3
**Recommendation:** 8
**Confidence:** 3

**Main Review:**

- Strength
  - This paper is well-written and easy to follow. The authors provide sufficient technical details for readers to understand and reproduce their work. Data, code, and the trained model will be open source.
  - This paper picks up an intriguing topic that aims to interpret deep learning models by investigating hidden units and summarizing their exemplar activation by natural language descriptions. The proposed method is concise, straightforward, and well-motivated.
  - The natural language descriptions can capture categorical, relational, and logical structure across different levels in the learned features. It’s nice to see low-level features like edges (“the top boundaries of horizontal objects”), middle-level features like shapes (“Poles and legs”), and relatively high-level features like objects (“dog faces”) could all be generated quite well by this same model.
  - The results suggest generalizability across different model architectures, datasets, and tasks. This makes MILAN readily useful for many other potential applications, including the three interesting experiments shown in section 5-7.
- Comments
  - The model is trained on a newly collected dataset MILANNOTATIONS. Each unit was annotated by three human participants. But the inter-annotator agreement among human annotations seems not quite high (Table 4, Figure 10), compared to the BERTScore between model-generated descriptions and human annotations (Table 2, Table 3). How did the authors handle this inter-annotator inconsistency during their model training? Is there any additional quality control/validation performed for this MILANNOTATIONS dataset?
  - Following the first point, I wonder if the authors would consider scaling up their methods by leveraging the existing large, multimodal datasets like GQA/Visual Genome or visual-language model trained on large paired image-text datasets like CLIP/ALIGN?
  - Fig. 3 is not referenced in the main text. And I think these failure modes are interesting, and taking a closer look at them might be inspiring for improving this model in future studies. Do authors have further comments or thoughts about this result?
  - The results in Fig. 4 are quite interesting. The bar chart suggests low-level visual features are more described by adjectives, middle-level units need more prepositions and verbs to describe relational features, and high-level units need more complex composition of words thus resulting in longer length and deeper parser trees. This probably aligns well with our intuition and expectation. For units that may contribute to those "non-robust" model behavior, are they described by more nouns with higher max word diff? Will the proposed MILAN model be able to detect those "non-robust" units and edit the network to improve its performance?
- Minor:
  - What do different dots refer to in Fig. 5?


**Summary Of The Paper:**

This paper describes a novel procedure (MILAN) to interpret deep learning models for computer vision by generating natural language description that specifies the activation selectivity of a given neuron in the model. For this aim, they first define an exemplar set of input image regions for each neuron by thresholding its activation value. Then they search a natural language description by optimizing the point-wise mutual information between descriptions and the exemplar set. The probability distributions for calculating the mutual information are approximated by training the SAT model and a two-layer LSTM language model on a newly collected dataset (MILANNOTATIONS), which includes annotations of 20k units labeled by human participants. The authors first test the generalizability of MILAN descriptions across different model architectures, datasets, and tasks, showing its privilege of generating higher agreement with human annotations compared to baseline methods. They then demonstrate three interesting applications of MILAN procedure and show how these natural language descriptions help us to understand and control the learned models.

**Summary Of The Review:**

Overall I think this work is well-motivated, technically sound, and showing promising results that support potential applications for interpreting and improving deep learning models for computer vision. Some minor changes could be made to improve the clarity. More details about how authors control / validate the quality of the MILANNOTATIONS dataset could be included.

---

> ### Author Response · Authors · 2021-11-16
> **Response to R1**
>
> Thank you for your comments! Answers to some of your questions follow.
>
> **Why is inter-annotator BERTScore lower than the model BERTScore?** The inter-annotator scores measure something fundamentally different than, e.g., the scores MILAN achieves on held out data. The lower inter-annotator scores reflect that different annotators often describe different aspects of the masked image regions; the higher held-out scores for MILAN reflect that MILAN is good at matching *at least one* of the human descriptions (since BERTScore is a multi-reference metric).
>
> **Can MILAN be scaled up by incorporating datasets like GQA/Visual Genome or large pretrained models like CLIP?** Absolutely! However, doing so will require solving several technical challenges, which we think is an excellent topic for future work.
>
> Datasets like GQA/Visual Genome, on their own, lack descriptions of low-level perceptual features like edges and colors. We collected MILANNOTATIONS to close that gap, but these other datasets could be used *in addition* to MILANNOTATIONS. For example, the bounding boxes in Visual Genome could be turned into masked image regions like the kind that MILAN is trained on, and the bounding box labels could be turned into free-form textual descriptions.
>
> Regarding large pretrained models like CLIP: these models could potentially provide better representations for the exemplar images that are input to MILAN. Since these models were trained on a wide range of images featuring many (object- and scene-level) visual concepts, leveraging their representations in the MILAN models might fill in gaps missing from MILANNOTATIONS. The challenge is that it is not immediately clear how to apply these models to *image regions* instead of entire images.
>
> **Figure 3 is not mentioned in the text; any further observations about the failure modes?** We’ll update the draft to include some discussion of Figure 3. We’ll also add more examples of model outputs.
>
> Figure 3 shows examples of MILAN’s broad failure modes: incorrect generalizations, vague descriptions, and specific semantic errors, including contextual mistakes. In some cases, MILAN generates correct captions that lack sufficient granularity: for instance, the sea life and sand in Figure 3 indeed share similar color patterns, but a more specific caption would mention sea cucumbers (unseen in the training examples). In other cases, MILAN makes semantic errors ranging from interpretable mistakes to, sometimes, vague or disfluent descriptions. Figure 3 shows two example semantic errors that we believe result from the difficulty of the captioning task, which requires using context to describe only a highlighted image region, and further requires generalizing across very different examples.
>
> **What do non-robust units look like? Can MILAN detect them automatically?** It is possible that the max-word-diff neurons (e.g. shown in Figure 12) reliably correspond to non-robust neurons of the kind identified by Mu & Andreas (2020). We are looking into whether this is the case, and will update the paper with our findings.

---

### Author Response · Authors · 2021-11-16
**General Response**

We would like to thank all the reviewers for their positive feedback! We are excited that they found our proposed method to be “concise, straightforward, and well-motivated” [R1] and the paper to have “clear hypotheses” [R3] with thorough testing [R2].

We also appreciate the many recommended improvements. **We are working on incorporating reviewer comments into an updated version of the draft, and we hope to have the updated version posted in the next day or two.** In the meantime, we wanted to respond to some of the comments.

---

### Public Comment · ~Iro_Laina1 · 2021-11-19
**A note on related work**

Dear authors,

Thank you for sharing your paper!  It was very interesting to read, and I am looking forward to the dataset.

As a related piece of work, I would like to  kindly note our own paper _“Quantifying Learnability and Describability of Visual Concepts Emerging in Representation Learning”_ published at NeurIPS 2020.

In the paper, we consider the problem of automatically generating natural language descriptions to characterize clusters of images forming _in the representation space_ of self-supervised algorithms, instead of individual neurons and their activating image patches. We also find that, often, image clusters correspond to compositions of concepts (e.g. dogs on grass) and that models trained on datasets such as Conceptual Captions are insufficient to capture sets of images that have lower-level details in common. Since in our work we did not have access to a specific dataset describing groups of images, I believe that MILANNOTATIONS can also find use in better analyzing, understanding, and evaluating concepts emerging in self-supervised representations, and I am thus looking forward to the data release.

As our paper tackles a related task, I would be grateful if you could discuss it in your updated paper.

Kind regards,
Iro Laina

---

> ### Author Response · Authors · 2021-11-19
> **Re: a note on related work**
>
> Thank you for bringing this to our attention! We agree that it's extremely relevant (and a cool paper!), and we'll discuss it in detail in the next version of our paper. We quite like the idea of captioning a group of images by selecting the centroid of their individual captions, and hope to incorporate it into some version of the baseline experiment suggested by reviewer YviU.

---

### Author Response · Authors · 2021-11-20
**Reviewer comments incorporated in newest revision**

Thanks again to all the reviewers for the helpful suggestions! We’ve uploaded a new version of the paper that incorporates some of the reviewer comments. These include:
- **[R3]** changing the title to “Natural Language Descriptions of Deep **Visual** Features” to better reflect the scope of the paper
- **[R1, R2]** mentioning Figure 3 in the main text and elaborating on MILAN’s failure modes
- **[R1, R2, R3]** adding a large set of randomly selected MILAN descriptions to the appendix, which highlights some additional success and failure cases of MILAN; see Figure 12
- **[R3]** adding dataset-wide corpus statistics for MILANNOTATIONS as an additional row in Table 5
- **[R3]** adding an additional table to the appendix with corpus statistics for MILAN-generated descriptions, and some discussions about how these compare to the MILANNOTATIONS corpus statistics
- **[R1]** clarifying how MILAN descriptions are sampled and reranked in Section 3.4; we did this in the text instead of adding a numbered algorithm
- **[R1]** a discussion in the appendix (Section D, Figure 14) about how the max-word-diff neurons found in Section 5 correspond to non-robust behavior in the model. Using one such neuron (which was automatically detected by MILAN), we constructed several copy-paste adversarial attacks for ResNet18-ImageNet that are similar to the attacks found in Compositional Explanations [Mu and Andreas, 2020].
- **[Iro Laina]** a discussion of “Quantifying Learnability and Describability of Visual Concepts Emerging in Representation Learning” in Related Work.

Finally, following **R3**’s suggestion, we experimented with using an off-the-shelf image captioning model to zero-shot describe a neuron’s top-activating image regions. We used a SAT model trained on the COCO captioning dataset. Because this captioning model is designed to take a single image as input, we feed it only the single most-activating image for each neuron, with the activation mask applied to the image at low opacity. This results, qualitatively, in vague captions that mention a single high-level object in the image and are frequently suffixed with the phrase “in the dark”--e.g., “a dog sits in the dark”--which both miss the interesting low-level visual properties of the image region and the visual properties it shares with the other top-activating image regions that were not input to the model. Quantitatively, this baseline achieves a BERTScore (f) of .15 relative to the human annotations in all of MILANNOTATIONS, which is substantially lower than both MILAN and NetDissect.

We are exploring whether this baseline can be improved by using the approach of Laina et al. (2020) to select a “most representative” description for the exemplar sets from among the SAT-generated captions. We will include any findings in the final version of the paper. However, both our initial results and Laina et al. suggest that this will not be sufficient to capture the full range of neuron behavior that we want to describe.

---

### Decision · Program_Chairs · 2022-01-20

**Decision:**

Accept (Oral)

**Comment:**

This paper presents a method to interpret neurons in the vision neural models by generating natural language description that specifies the activation selectivity of a given neuron. The proposed method first identifies an exemplar set of input image regions that corresponds to a neuron, then searches a natural language description by optimizing the point-wise mutual information between descriptions and the exemplar set.

Strength:
- Reasonable method design and clear writing
- Important problem and broad applications
- Extensive experiments for evaluation of the proposed method

Weakness:
- Need more discussion on the limitations of the proposed method
- Elaboration on the human inter-annotation agreement
- Analysis on method transferability across tasks.